# Sparsity and Superposition in Mixture of Experts

## Abstract

Mixture of Experts (MoE) models have become central to scaling large language models, yet their mechanistic differences from dense networks remain poorly understood. Previous work has explored how dense models use *superposition* to represent more features than dimensions, and how superposition is a function of feature sparsity and feature importance. MoE models cannot be explained mechanistically through the same lens. We find that neither feature sparsity nor feature importance cause discontinuous phase changes, and that network sparsity (the ratio of active to total experts) better characterizes MoEs. We develop new metrics for measuring superposition across experts. Our findings demonstrate that models with greater network sparsity exhibit greater *monosemanticity*. We propose a new definition of expert specialization based on monosemantic feature representation rather than load balancing, showing that experts naturally organize around coherent feature combinations when initialized appropriately. These results suggest that network sparsity in MoEs may enable more interpretable models without sacrificing performance, challenging the common assumption that interpretability and capability are fundamentally at odds.

## 1 Introduction

Mixture of Experts (MoEs) have become prevalent in state-of-the-art language models, such as Qwen3, Mixtral, and Gemini (Yang et al., 2025a; Jiang et al., 2024; Google DeepMind, 2025), primarily for their computational efficiency and performance gains (Shazeer et al., 2017; Fedus et al., 2022). Subsequent work improve routing (e.g., Expert-Choice routing) and training stability/transfer (e.g., ST-MoE) (Zhou et al., 2022; Zoph et al., 2022). Theoretical and empirical results further show that learnable routers can discover latent cluster structure in data, providing insight for why experts specialize (Dikkala et al., 2023). However, despite their widespread adoption, MoEs remain poorly understood from a mechanistic interpretability perspective.

Interpretability-oriented approaches have sought to make expert behavior more transparent. Yang et al. (2025b) proposes MoE-X, which encourages sparsity-aware routing and uses wide, ReLU-based experts to reduce polysemanticity. Park et al. (2025) introduce Monet, scaling the number of experts to enable capability editing via expert activation. Yet these works largely focus on architectural changes; we still lack a mechanistic understanding of how MoEs represent features, how experts affect superposition, and whether experts naturally specialize without extra regularization. Mu & Lin (2025) survey MoE research and identify mechanistic interpretability as a key open challenge.

A fundamental challenge in interpreting neural networks is the phenomenon of superposition: when models represent more features than they have dimensions. This allows networks to pack many sparse features into fewer neurons at the cost of making individual neurons polysemantic and difficult to interpret.

MoE architectures introduce a new dimension to this problem: network sparsity. Unlike dense models that activate all neurons regardless of input, MoEs activate a fraction of their total parameters (Shazeer et al., 2017). While dense models exploit feature sparsity by

packing many sparse features into shared neurons, MoEs can afford to be more selective, potentially dedicating entire experts to specific feature combinations.

We investigate whether (1) MoEs exhibit less superposition than their dense counterparts, (2) there is a discrete phase change in the amount of superposition of a particular feature in MoE experts across its relative importance and overall feature sparsity, as seen in dense models, and (3) we can understand expert specialization through the lens of feature representation rather than just load balancing.

We explore these questions using simple models that extend Elhage et al. (2022)'s framework to MoEs. Our key contributions are as follows: (1) unlike dense models, MoEs do not exhibit sharp phase changes, instead showing more continuous transitions as network sparsity increases; and (2) MoEs consistently exhibit greater monosemanticity (less superposition) than dense models with equivalent active and total parameters, with individual experts representing features more cleanly; (3) we propose an interpretability-focused definition of expert specialization based on monosemantic feature representation, showing that experts naturally organize around coherent feature combinations rather than arbitrary load balancing.

## 2 Related Work

**Superposition and Feature Representations.** The Linear Representation Hypothesis suggests networks represent concepts as directions in activation space (Park et al., 2024), yet the number of interpretable features often exceeds the available dimensions. Elhage et al. (2022) formalized this phenomenon as superposition, demonstrating that dense models rely on non-orthogonal feature packing to maximize capacity at the cost of polysemanticity. While superposition is theoretically efficient (Scherlis et al., 2025), it necessitates complex post-hoc disentanglement methods, such as Sparse Autoencoders, to recover monosemantic features (Bricken et al., 2023). Recent work has examined how data correlations shape superposition (Prieto et al., 2025), and how interference patterns emerge and can be mitigated (Gurnee et al., 2023). Our work extends this line of inquiry to MoE architectures, demonstrating that network sparsity—rather than feature sparsity alone—governs representational strategies.

**Interpretability of MoEs.** While MoEs have become the standard for scaling large language models (Shazeer et al., 2017; Fedus et al., 2022; Jiang et al., 2024), mechanistic understanding of their internal representations lags behind their dense counterparts. Existing analysis largely focuses on macroscopic behaviors, such as routing stability (Zoph et al., 2022), expert choice statistics (Zhou et al., 2022), or latent cluster discovery (Dikkala et al., 2023). More recent interpretability-focused approaches attempt to force specialization through architectural constraints, such as predefined concept routing (Yang et al., 2025b) or scaling expert counts to match vocabulary sizes (Park et al., 2025). However, these approaches often prioritize capability editing over explaining intrinsic feature geometry. We address this gap by analyzing how experts affect superposition, demonstrating that MoEs exhibit greater monosemanticity than dense models and proposing a feature-based definition of expert specialization.

## 3 Background

A primary focus of Mechanistic Interpretability is to reverse engineer neural networks; one method is to decompose model representations into a set of human interpretable concepts named 'features'. These features are often assumed to be linear, that is, any hidden state $h$ can be described as

$$h = \sum_{i \in F} \alpha_i \vec{f}_i + \vec{b}$$

where $\vec{f}_i$ is the direction corresponding to feature $i$, $\alpha_i$ is the activation strength of this feature (roughly, the degree to which feature $i$ is present in the input), and $F$ is the set of all represented features.

The *superposition hypothesis* contends that models are capable of representing far more features than dimensions, i.e. $|F| > m$ for $h \in \mathbb{R}^m$ (Elhage et al., 2022). In order to have many more features than dimensions in a latent space, features vectors in $F$ must be packed such that they are not all orthogonal. When two features have interference $\langle \vec{f_i}, \vec{f_j} \rangle \neq 0$, we say they are in *superposition*. This superposition is acceptable so long as features are sparsely active (a characteristic of most online text data), though it comes at the cost of interpretability, as $\alpha_i$ contains spurious activations unrelated to feature $i$ being present in the input.

Monosemanticity is a characteristic of individual neurons, where a neuron's activation cleanly corresponds with a single $\alpha_i$ (i.e., features are basis-aligned). When features are orthogonal (not in superposition) but not basis-aligned, neurons remain *polysemantic* even though feature interference is minimal. In this paper, we focus on reducing superposition rather than enforcing basis-alignment. For brevity, when we describe models, experts, or features as "more monosemantic," we mean they display less superposition.

## 4 DEMONSTRATING SUPERPOSITION

MoEs are often conceptualized as compositions of dense models, where each expert behaves like an independent dense network. However, whether experts actually represent features similarly to dense models remains unclear. We investigated this by comparing how MoEs and dense models differ in superposition.

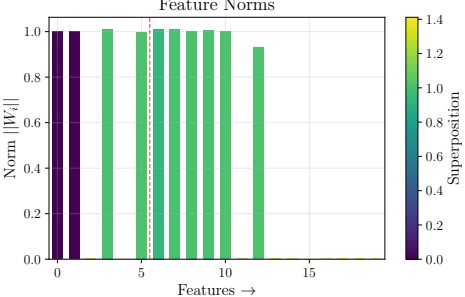

(a) Norm of each feature's weight vector $\|W_i\|$, with colors indicating superposition status (green for features in superposition, purple for monosemantic features).

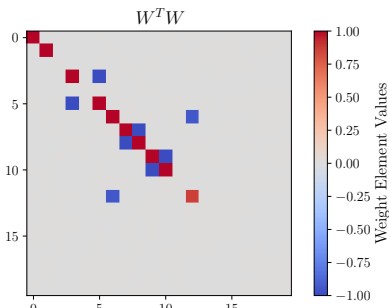

(b) $W^\top W$ matrix where each cell represents $(\hat{W}_i \cdot W_j)$, revealing interference patterns between features.

Figure 1: Feature representation and superposition in a dense model with $n = 20$ features and $m = 6$ hidden dimensions, with importance $I = 0.7^i$ and uniform feature density $(1 - S) = 0.1$. Superposition (color) is given by $\sum_j (\hat{W}_i \cdot W_j)^2$.

### 4.1 EXPERIMENTAL SETUP

Our goal is to explore how a MoE can project a high-dimensional vector, $x \in \mathbb{R}^n$ into lower-dimensional expert representations, $h \in \mathbb{R}^m$ and then accurately recover it. This extends the framework of Elhage et al. (2022) to the MoE setting.

**The input distribution.** The input vector $x$ represents the activations of an idealized, disentangled model, where each dimension $x_i$ corresponds to a distinct, independent feature—effectively, a perfectly neuron-aligned and monosemantic representation. We take $x_i \sim U(0, 1)$, except for a given sparsity $S \in [0, 1)$, $P(x_i = 0) = S$. Not all features contribute equally to the loss. To model that features vary in utility, we assign each feature $x_i$ a scalar importance $I_i$ which weights the reconstruction loss. Additionally, to isolate and study phase transitions for a single feature, we scale the magnitude of the last feature by a factor $r \in \mathbb{R}^+$, such that $I = (1, 1, ...r)$. Thus, $I_i$ and $r$ vary the signal strength of the features, allowing us to test expert sensitivity to feature magnitude.

**Model Architecture.** The MoE consists of $E$ experts, where each expert $e$ is parameterized by a weight matrix $W^e \in \mathbb{R}^{m \times n}$ and a bias $b^e \in \mathbb{R}^n$. Inputs are assigned to the top-$k$ experts via a learned router $g(x) = \text{softmax}(W^r x)$ where $W^r \in \mathbb{R}^{E \times n}$. Each active expert projects the input to a lower-dimensional hidden state $h^e = W^e x$ and generates a reconstruction $\hat{x}^e = \text{ReLU}((W^e)^\top h^e + b^e)$ [1]. The final output is the weighted sum of the active experts: $x' = \sum_{e \in \text{top-}k} w_e \hat{x}^e$ where $w_e$ are the renormalized gating weights.

We train our models with an L2 reconstruction loss weighted by feature importances, $I_i$ given by $\mathcal{L} = \sum_x \sum_i I_i (x_i - x'_i)^2$. In Section 4, to prevent expert collapse, we add the standard auxiliary load balancing loss (Fedus et al., 2022), defined as $\mathcal{L}_{\text{aux}} = \alpha N \sum_{e=1}^N f_e P_e$. Here, $N$ is the total number of experts, $f_e$ is the fraction of samples in a batch routed to expert $e$, $P_e$ is the average gating probability assigned to expert $e$ across the batch, and $\alpha = 0.01$ controlling the penalty strength. We deliberately omit auxiliary load balancing in Section 4 to isolate the intrinsic architectural bias of the MoE regarding superposition.

## 4.2 Measuring feature capacity

To analyze feature representations across architectures, we compared two fundamental properties of the features: *representation strength* and *interference* with other features. We measured the norm of a feature weight vector in an expert $e$ given by $\|W_i^e\|$. It represents the extent to which a feature is represented within the expert $e$. $\|W_i^e\| \approx 1$ if feature $i$ is fully represented in expert $e$ and zero if it is not learned. We visualize the interference of a feature $i$ with other features in expert $e$ using the Gram matrix $W^\top W$, where off-diagonal elements represent pairwise interference.

As shown in Figures 1a and 2a, the dense and the MoE represent a comparable number of features (10 vs 8) with similar norms for equal total parameters ($m_{\text{dense}} = \sum m_{\text{experts}} = 6$). While the MoE experts exhibit some local superposition (e.g., Expert 0 in Fig 2b), the global interference structure is strictly partitioned. Unlike the dense model (Figure 1b, where any feature can interfere with any other, the MoE enforces a block-diagonal structure where features routed to different experts have zero interference. This demonstrates that MoEs allocate representational capacity by partitioning the feature space, reducing the global scope of interference.

**Expert Feature Dimensionality.** We want to understand how MoEs allocate their limited representation capacity differently from the dense model. We measured feature dimensionality, which represents the "fraction of a dimension" that a specific feature gets in a model (Elhage et al., 2022). For a feature $i$, we define its dimensionality in expert $e$ by

$$D_i^e = \frac{\|W_i^e\|^2}{\sum_j \left(\hat{W}_i^e \cdot W_j^e\right)^2} \tag{1}$$

$D_i^e$ is bounded between zero (not learned) and one (monosemantic). The total capacity for a MoE can thus be defined as $D = \sum_e \sum_{i=1}^n D_i^e$.

**Efficient Packing.** When the features are "efficiently packed" in a model's representation space, the dimensionality of all the features add up to the number of embedding dimensions, i.e. $\sum_{i=1}^n D_i^e \approx m$ (Cohen et al., 2014; Scherlis et al., 2025; Elhage et al., 2022). In the case of a MoE, the relation becomes $\sum_e \sum_{i=1}^n D_i^e \approx E \cdot m$. Empirically, we find that both dense and MoE models satisfy the above dimensionality constraint, meaning that MoEs achieve the same efficiency in packing features as the dense models for the same total parameters.

**Features per Dimension & Network Sparsity.** Since both dense and MoE models "efficiently pack" features in their representation space, we compared the differences in

---

[1] The ReLU at the output ensures non-negative reconstructions, which matches our input distribution where $x_i \in [0, 1]$ when nonzero. Furthermore, the off-diagonal terms in $(W^e)^\top W^e$ create negative interference and ReLU suppresses these negative components. When features are sparse, negative interference becomes effectively "free" as it is filtered to zero, incentivizing model configurations with negative off-diagonal terms (e.g., antipodal pairs).

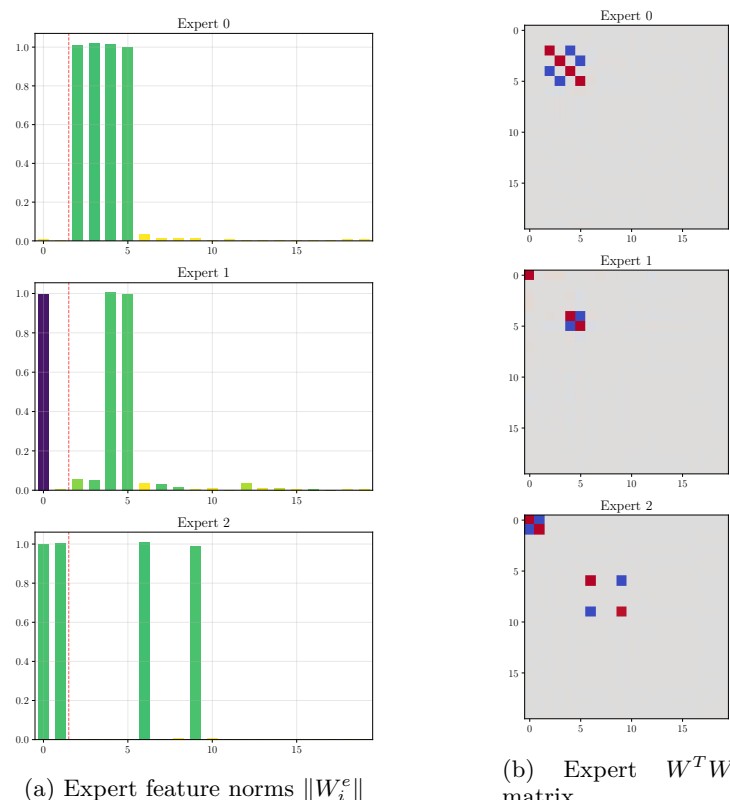

(a) Expert feature norms $\|W_i^e\|$

(b) Expert $W^T W$ matrix

Figure 2: Feature representation and superposition in a MoE with $n = 20$ features, 3 total experts, and $m = 2$ hidden dimensions per expert (top-$k = 1$ routing), with importance $I = 0.7^i$ and feature density $1 - S = 0.1$.

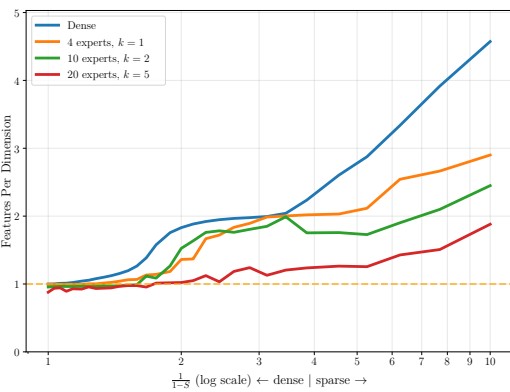

Figure 3: Features per dimension versus inverse feature density ($\frac{1}{1-S}$) for dense and MoE architectures with uniform feature importance ($I_i = 1.0$). The dense model ($n = 100$, $m = 20$) has the most superposition, which decreases with increasing expert count: 4 experts with $m = 5$, $k = 1$ (orange); 10 experts with $m = 2$, $k = 2$ (green); 20 experts with $m = 1$, $k = 5$ (red). All models have equal total parameters and similar $k/E$. The dashed line at 1.0 marks monosemantic representation.

number of features per dimension across the models. This allowed us to exactly measure superposition in both models and how the number of experts in a MoE affects superposition for different feature sparsities. If the features per dimension is greater than one, then the

features are in superposition since the model is representing more features than there are dimensions. We define features per dimension for a MoE by

$$\frac{1}{k}\sum_{e=1}^{E} p_e \frac{\|W^e\|_F^2}{m} \tag{2}$$

where $\|W\|_F^2$ is the Frobenius norm and $p_e$ is the expected probability that expert $e$ is used across a batch of input samples, i.e. the average renormalized gating weight after top-$k$ routing.

For MoEs with *equal* number of total parameters as the dense model, we observe that the dense model has a higher number of features per dimension (Figure 3), i.e. more superposition. This indicates that the dense model utilizes superposition to represent a greater total number of features ($n_{\text{learned}} > m_{\text{total}}$), whereas the MoE tends to cap its representation at the monosemantic limit ($n_{\text{learned}} \approx m_{\text{total}}$). Furthermore, as we increase the total number of experts in the MoE—keeping the total parameters and the ratio $k/E$ roughly the same—the number of features per dimension decreases or alternatively has less superposition. *The greater the number of experts, the less superposition in the model.* Concretely, features become more monosemantic with increasing number of experts. Furthermore, more superposition in the dense model allows it to achieve consistently lower reconstruction loss compared to the MoEs as shown in Figure 6 in Appendix A.1 with difference in loss at any given sparsity of $\sim 0.03 - 0.08$. But as the number of experts increases, the MoEs achieve consistently comparable loss to the dense models. See Appendix A.2 for a theoretical intuition.

## 5 Phase Change

Although MoEs and dense models learn a similar number of features, MoEs distribute them across experts with less interference. This suggests that network sparsity reshapes how features are allocated rather than how many are learned. We examined how properties of the input distribution—such as feature sparsity and importance—drive this allocation and whether they induce physics-inspired *phase changes* in representation.

Models have a finite way of representing features; each feature may be ignored, superimposed, or monosemantic. Phase change is the observation that sometimes there are discrete boundaries between regions, which are functions of feature sparsity and relative importance.

Dense toy models exhibit discontinuous 'phase changes' between internal feature representations (Elhage et al., 2022). By varying the sparsity and relative importance of features in the input distribution, we can elicit different behavior; for example, more feature sparsity encourages greater superposition. Analyzing the phase diagram of each expert in MoEs demonstrates they employ different representational strategies compared to dense models.

**Setup.** We follow the same setup as Section 4.1, except with load balancing loss to avoid specialization collapse—when experts and routers fall into local minima where certain experts are entirely ignored—in certain setups. There are three models setups, all with one active expert ($k = 1$): (A) n=2, m=1; (B) n=3, m=1; and (C) n=3, m=2. We report the expert-specific phase diagram across feature sparsity and last-feature relative importance for varying network sparsity by increasing the experts ($E$) up to the number of input feature dimensions ($n$).

In this section we fix active parameters rather than total parameters such that $m_{dense} = km_{moe}$ (ignoring router parameters). The reason is to compare within model architectures; otherwise, any observed differences could be attributed to architectural changes instead of the number of active parameters. Coincidentally, 4.C.1/1 has the same number of active parameters as 4.B.X/2. But the latter has only one hidden dimension ($m = 1$) to encode the same number of input features ($n = 3$) as the former, with two hidden dimensions ($m = 2$), making it difficult to use superposition to understand specialization.

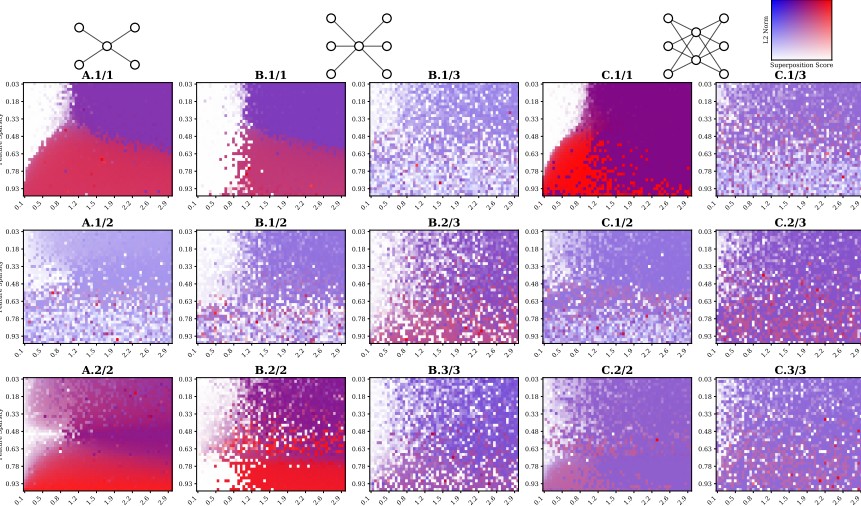

Figure 4: For a particular expert and input dimension (feature), we can decode how it is embedded in the hidden dimension—whether it is ignored (white), monosemantic (blue-purple), or superimposed (red). We plot joint feature norm ($||W_n||^2$) and superposition score ($\sum_{j<n}(\hat{W}_n \cdot W_j)^2$) across varying feature sparsity $S \in [0.1, 1]$ and relative last feature importance $I_n \in [0.1, 3]$, where the subscript $_n$ denotes the last feature of $n$ total features. For each cell, we train ten models and select the one with the lowest loss. We used load balancing loss in this section. We plot joint feature norm and superposition for the last feature: low L2 norm ($||W_n||$) is white, denoting the model is ignoring the last feature; otherwise a low superposition score is blue-purple to indicate monosemantic representation of the last feature. Red indicates the feature is represented in superposition. Cell $(i, j)$ in subfigure X.e/E denotes the expert e of E total experts trained on architecture X for last feature importance $I_n = i$ and sparsity $S = j$; X.1/1 indicates a dense model.

**Results and Takeaways.** In all single-expert (dense) cases, we observed a clear phase change (Figure 4.X.1/1), affirming the work of Elhage et al. (2022). When we increased the total number of experts, discrete phase changes disappeared. Some experts in MoEs with $E = 2$ are reminiscent of their respective dense cases (Figure 4.X.2/2), but exhibit more continuous transitions. In each case, one expert became more monosemantic, specializing in the most important feature by relative importance. Experts dissimilar from the dense cases universally have much lower superposition scores (they are bluer), indicating more monosemantic representations. This aligns with the conclusions of the previous section—MoEs favor lower superposition scores compared to their dense counterparts.

For the $n = 2, m = 1$ setup (Figure 4.A), the dense model does not represent the last feature when feature sparsity is low. However, the comparable MoE model preserves the last feature much more because it has the capacity. With three input dimensions (Figure 4.B), the MoEs do not exhibit this behavior because the experts are superimposing the other two features; there is no space for the third feature within one hidden dimension. Unlike the other two cases, for architecture B the hidden dimension with superposition is not sufficient, in the high-sparsity regime, to represent all features. Yet we do not see clear phase change—except for the $0.5 - 0.7$ feature sparsity region in 4.B.2/2, where it is mostly discrete but mixed. For $m = 2$ the white region in the dense model (Figure 4.C.1/1) (in the mid- to low-feature sparsity domain, when the feature is relatively less importance than the others) ignores the last feature. However, as network sparsity increases—across all other Figure 4.C—the models represent the last feature with greater L2 magnitude ($||W_3^1|| < ||W_3^2|| < ||W_3^3||$). In other words, the dimensionality in the low relative-importance region increased with increasing network sparsity, as demonstrated in Figure 3.

We observed a window of feature sparsity from roughly 0.48 to 0.7 in Figures 4.B.2/2, 4.C.1/2, and 4.C.2/2 where there is heavy mix of polysemanticity, monosemanticity, or

ignorance. This indicates there is a middleground in MoEs with comparable loss between polysemantic and monosemantic representations which make it difficult to consistently commit to the strategies we observe in low and high feature sparsity domains. We see no such pattern in dense models—evidence that MoEs learn different representational strategies.

**Conclusion.** These experiments are all top-$k = 1$, so only one expert is active at a time. Even so, we see vastly different behavior even the in $E = 2$ case, including when the hidden dimension capacity with superposition is sufficient to represent all features. This leads us to conclude it is misleading to think of MoEs as an aggregation of dense models. The mechanism of the router which allows experts to observe only a subset of the feature domain vastly modifies the behavior and learning of the experts.

## 6 EXPERT SPECIALIZATION

Since MoEs exhibit less superposition, we now examine the organization of such monosemantic features within experts and its relation to specialization.

Expert specialization in MoEs traditionally centers around load balancing between experts across all inputs (Chaudhari et al., 2025). However, this fails to capture the natural intuition of specialization, wherein an expert is only used when appropriate concepts—those the expert is specialized in—are present in the input.

We define an expert as specialized if it *occupies* certain feature directions in the input space, and if it represents said features relatively monosemantically. We demonstrate that these two conditions are directly correlated, and show how the presence of these two conditions encourages load balancing across experts.

Because we fix $k = 1$, the feature space is partitioned into convex cone regions (see Appendix A.3), with each region routed to a particular expert. By definition, this means $\forall s > 0, \ \vec{x} \in C \rightarrow s\vec{x} \in C$, where $C$ is the set of points contained within the cone and $s$ is any positive scalar. If a particular feature vector $\vec{x}$ is routed to an expert, then all $s\vec{x}$ are routed to that same expert. In this case, we say that feature $x$ is contained within expert $e$, and as such expert $e$ *occupies* $x$.

We empirically find that small models that distribute the input space across more experts tend to achieve lower loss (see Appendix A.4). This warrants a question: does the allocation of the input space to certain experts imply any characteristics regarding those experts? Our definition of expert specialization suggests that this allocation implies *monosemanticity*, which we will see is a correlation that holds for larger toy models (e.g., $m = 10$).

In models with $n > 2$, we explore whether initializing experts to occupy features in the input space cause the experts to be more monosemantic w.r.t. those features. Separately, we see if, for the features an expert has chosen to represent monosemantically, the expert occupies those features in the input space.

When the gate matrix is initialized with ones along the main diagonal, each expert monosemantically represents the single feature it initially occupied, and only that feature, as shown in Figure 5a. When the router is ordered k-hot initialized, the first expert monosemantically represents four of the five features it initially occupied, as shown in Figure 5b. The other experts, initialized over other features, did not monosemantically represent these less important features, nor did they monosemantically represent the five most important features they were not initialized over. When we break the ordering of feature importance and randomize the features each expert initially occupies, each expert monosemantically represented only the most important feature(s) it was initialized over, as shown in Figure 5c.

There is a strong correlation between the features that are initially routed to an expert and which features that expert represents monosemantically. Furthermore, we observe that experts only monosemantically represent important features. This is true if we initialize each expert with one important feature explicitly, or if we give it a set of features, upon which it selects the most important feature itself and represents it monosemantically.

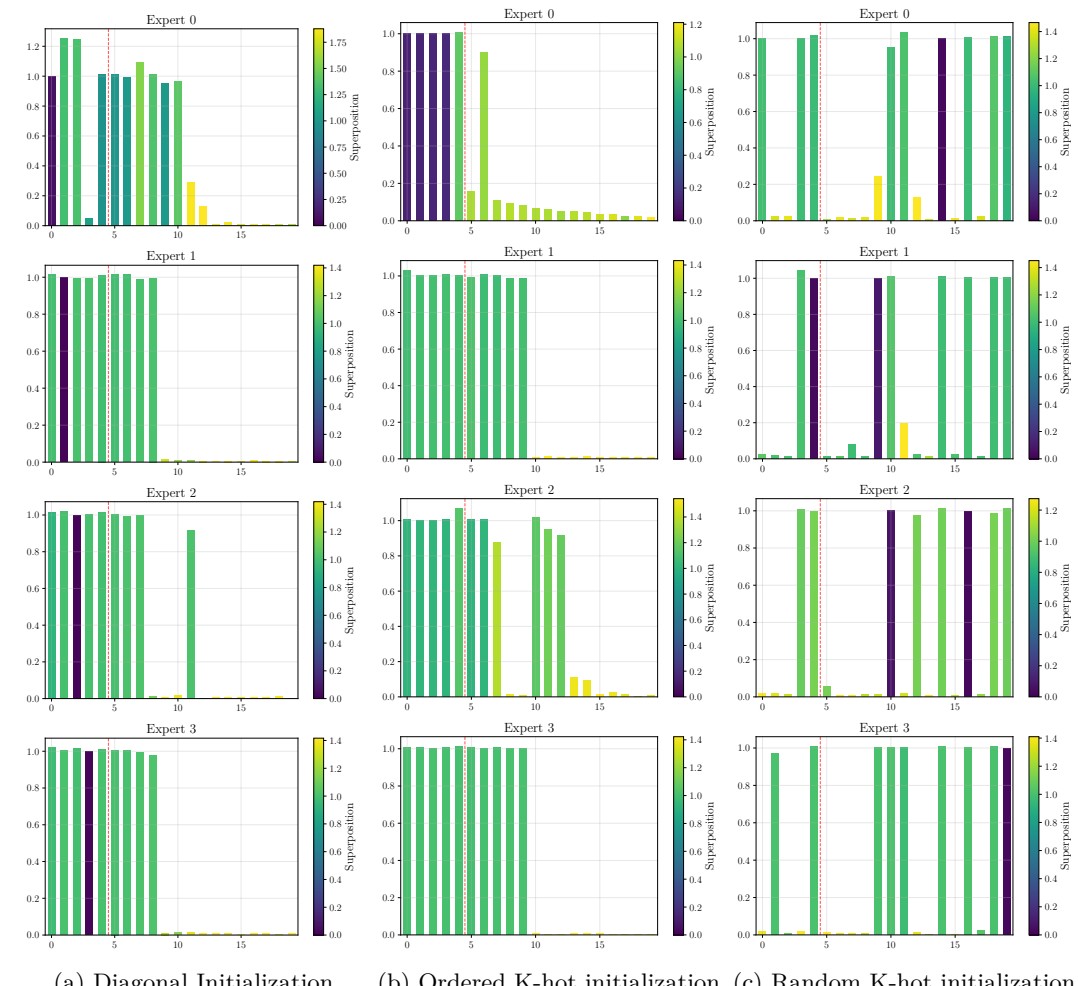

(a) Diagonal Initialization  (b) Ordered K-hot initialization  (c) Random K-hot initialization

Figure 5: Expert feature norms $||W_i^{(e)}||$ and superposition (color) results for three different initialization schemes, with $n = 20, m = 5, E = 4, S = 0.1$. In **(a)**, the gate matrix is initialized along the main diagonal ($W_i^r = \hat{e}_i$, the basis vector for that dimension), and relative feature importance decreases exponentially in order from feature one to 20. In **(b)**, the gate matrix is initialized to an "ordered k-hot", such that the first expert aligns with the first five features, and each subsequent expert aligns with the next five features. Relative feature importance is the same as **(a)**. In **(c)**, the gate matrix is initialized to a "random k-hot", where each expert is assigned five random features such that experts share no common feature but cover all 20 features collectively. Relative feature importance decreases exponentially but is randomly distributed across features.

In the case of uniform feature importance, experts place all features in superposition with higher levels of interference. Despite this, the features an expert initially occupies are still *relatively* more monosemantic, on average achieving superposition scores 1.03 standard deviations below the average for that expert.

We investigated whether there is a correlation between experts representing certain features monosemantically, and said experts occupying those features in the input. To do this, we measure usage statistics when those features are *one of many* active features, and when they are the *only* active features. This second case is equivalent to measuring the probability that the expert occupies these features. The correlation holds both in xavier and k-hot initialization schemes, as seen in Table 1. Given $E = 10$, a mean expert usage of ~10% indicates an even load balancing across experts. In all cases, when the corresponding monosemantic

feature(s) for an expert is active, the usage of the expert increases significantly. When this feature(s) is the only active feature, the expert dominates the usage. In the k-hot initialization scheme, *100%* of all features monosemantically represented by an expert are occupied by that same expert.

Table 1: Monosemantic feature and usage statistics per expert for $n = 100, m = 10, E = 10$. One hundred models are trained for each initialization scheme (xavier and k-hot), providing 1000 experts in total for each. Each statistic is aggregated across models, classifying experts based on the number of features they represent monosemantically. For the feature(s) an expert represents monosemantically, we track the expert usage when said feature(s) is one of several active features in the input, as well as the expert usage when said feature(s) is the *only* active feature in the input.

| Xavier Initialization | | | | |
|---|---|---|---|---|
| Number of monosemantic features per expert | Number of experts (out of 1000) | Mean expert usage (%) | Mean expert usage; feature(s) active (%) | Mean expert usage; only feature(s) active (%) |
| 0 | 461 | – | – | – |
| 1 | 387 | 9.595 | 17.94 | 67.18 |
| 2 | 138 | 9.599 | 30.29 | 95.65 |
| 3 | 13 | 8.363 | 40.19 | 100.0 |
| 4 | 1 | 1.428 | 14.69 | 100.0 |
| 5 | 0 | – | – | – |
| K-Hot Initialization | | | | |
| Number of monosemantic features per expert | Number of experts (out of 1000) | Mean expert usage (%) | Mean expert usage feature(s) active (%) | Mean expert usage only feature(s) active (%) |
| 0 | 335 | – | – | – |
| 1 | 382 | 10.00 | 23.94 | 100.0 |
| 2 | 227 | 10.02 | 46.61 | 100.0 |
| 3 | 47 | 10.09 | 62.00 | 100.0 |
| 4 | 8 | 9.95 | 70.30 | 100.0 |
| 5 | 1 | 9.62 | 74.79 | 100.0 |

As experts represent more features monosemantically, they can be seen as more specialized. Their usage on arbitrary input decreases, but conditional on their specialized features being active, their usage increases far greater than other experts. This holds true for all cases except the xavier initialized model with a four monosemantic feature expert, where there is a significant drop in utilization.

## 7 CONCLUSION

We investigated how experts affect superposition in MoEs, showing that MoEs consistently exhibit greater monosemanticity than dense networks while not exhibiting a phase change. We proposed a feature-based definition of expert specialization, demonstrating that experts naturally organize around coherent features when initialization encourages this specialization. However, our findings are based on simple autoencoder toy models with synthetic data, leaving open questions about generalization to large-scale transformers where the feature distribution is unknown (see Appendix A.6). Despite these limitations, we show how toy MoEs achieve comparable loss while maintaining more interpretable representations—challenging the prevalent zeitgeist that mechanistic interpretability and model capability are fundamentally in tension. Future work should explore what favors monosemanticity in MoEs, how training dynamics of MoEs differ from those of the dense model, and when specialization emerges. Answering these questions can inform the design of more interpretable, high-performing language models.

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

# A APPENDIX

## A.1 MEASURING LOSS FOR VARYING SPARSITY & EXPERTS

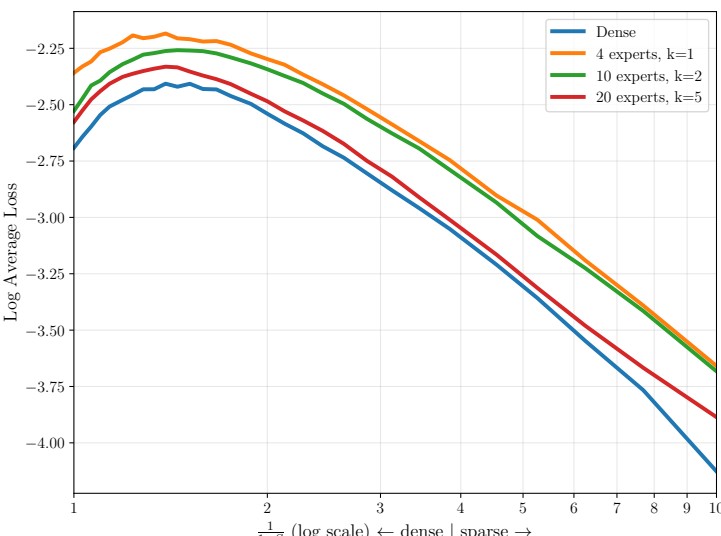

Figure 6: Log average loss versus feature density ($\frac{1}{1-S}$) for dense ($m = 20$) and MoE (4 experts, $k = 1$, $m = 5$), MoE (10 experts, $k = 2$, $m = 2$), and MoE (20 experts, $k = 5$, $m = 1$) models, all with uniform feature importance ($I_i = 1.0$) for $n = 100$ input features. Results are averaged over five runs per sparsity level. Although dense model outperforms all MoEs at every sparsity level, as the number of experts increases, the MoE loss gets closer to the dense model.

## A.2 THEORETICAL INTUITION FOR SUPERPOSITION

In dense models, superposition emerges to exploit the gap between sparse features and dense computation, compressing rarely active features into shared dimensions. MoEs, however, "eat this same gap" structurally through conditional computation. By aligning activation sparsity with feature sparsity, MoEs remove the computational penalty for having dedicated, rarely-activating neurons. As noted by Elhage et al. (2022), when a model only expends computation on active features, splitting polysemantic neurons into dedicated monosemantic ones becomes the optimal strategy, effectively trading the *compression* of superposition for the *selection* of routing.

The structural change manifests geometrically as a reduction in *interference*. While the global ratio of number of features represented to parameters remains constant, the router effectively partitions the feature space, ensuring that a feature routed to expert $e$ only competes for capacity with the subset of features also assigned to that expert. Consequently, interference is governed by the local expert matrix $(W^e)^\top W^e$ rather than the global $W^\top W$ of a dense model. This partitioning drastically reduces the number of interfering features for any given feature vector, minimizing the optimization pressure to pack features in superposition and allowing experts to learn monosemantic representations.

## A.3 ROUTER SUBSPACES ARE CONVEX CONES

In the regime of $k = 1$, the router function $g(x) = \text{softmax}(W^r x)$ is equivalent to $\text{argmax}(W^r x)$. The region routed to expert $i$ can be represented as $\forall j \neq i, (w_i - w_j)^\top x > 0$ where $w_i$ and $w_j$ are row vectors of $W^r$. This is a homogeneous linear inequality. Regions bounded by such inequalities are by definition convex cones. If a particular $x$ satisfies this inequality, then multiplying both sides by any positive scalar $s$ will still satisfy the inequal-

ity. Furthermore, if $x_1$ and $x_2$ satisfy this inequality, then any $x = x_1\lambda + (1 - \lambda)x_2$ for $\lambda \in [0, 1]$ will also satisfy the inequality.

In the case of $k > 1$, the region of inputs which get sent to a particular expert $e$ becomes a union of convex cones. Generally, the union of a convex cone is not itself a convex cone. Therefore, the understanding of experts occupying feature directions may not hold beyond $k = 1$.

## A.4 EXPERT ROUTING WITH DIFFERENT INITIALIZATION SCHEMES

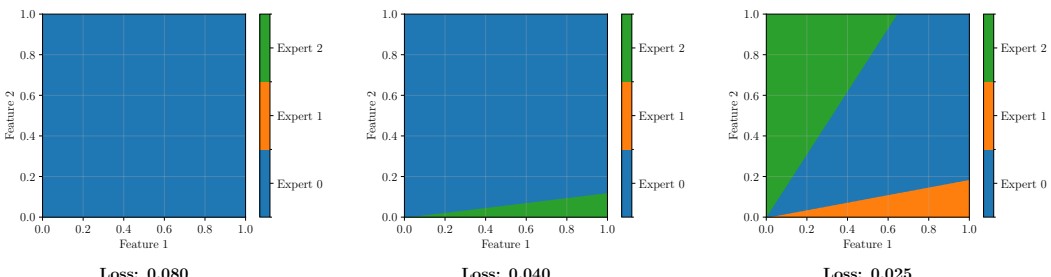

Figure 7: Expert routing of three identical models with differing initialization schemes. We use $n = 2$, $E = 3$, $m = 1$. The first model (left) has the worst performance (loss: 0.08) and routes all inputs to one expert. The second model (middle) has better performance (loss: 0.04) and routes a small portion of inputs, specifically those when feature 1 is active, to a second expert. The third model (right) has the lowest loss (loss: 0.025), and distributes the input space among all experts. One expert is chosen when only feature 1 is active, one when only feature 2 is active, and one when both are active.

In small models ($n = 2, m = 1, E > 1$), we empirically find that models that distribute the input space across more experts tend to achieve lower loss, testing with $E \in [2, 7]$. Holding $n = 2$ allows us to visualize which portions of the input space get routed to which experts, as seen in Figure 7.

## A.5 ANALYTIC MODEL EQUIVARIANCE

For the toy setup of single-layer, single-nonlinearity, top-$k = 1$ MoEs, there exists a theoretical map between any dense model and a monosemantic MoE with an equivalent number of active features under a sparsity constraint.

Assume there exists an upper bound for the number of active features $a$ for any input such that $\forall x \in D : |\{i : x_i \neq 0\}| \leq a$. Furthermore, assume that $a$ is no greater than the hidden dimensionality, $m$, of an expert, providing an upper bound on the number of features a model has to represent. Assume also that the hidden dimensions is smaller than the total number of input features $n$ ($a \leq m \leq n$). To construct the monosemantic MoE, for each possible subset $S \subseteq \{1, 2, \ldots, n\}$ with $|S| \leq a$—meaning the size of the subset of active features is smaller than or equal to $a$—create an expert which monosemantically preserves those features. (In fact, you can take only the subsets such that $|S| = a$.) The router then selects the expert which corresponds to those active features (of which there will never be more than $a$, by assumption):

$$\text{Router}(x) = \arg\max_S \mathbb{I}[\text{support}(x) = S]$$

where $\text{support}(x) = \{i : x_i \neq 0\}$. Since $|S| \leq a \leq m$, each expert has sufficient capacity to represent its assigned features without superposition. To reiterate, only $a$ features are active and every unique combination of active features receives its own dedicated expert with sufficient capacity to represent those features monosemantically. So, the number of possible experts needed is $\binom{n}{m}$.

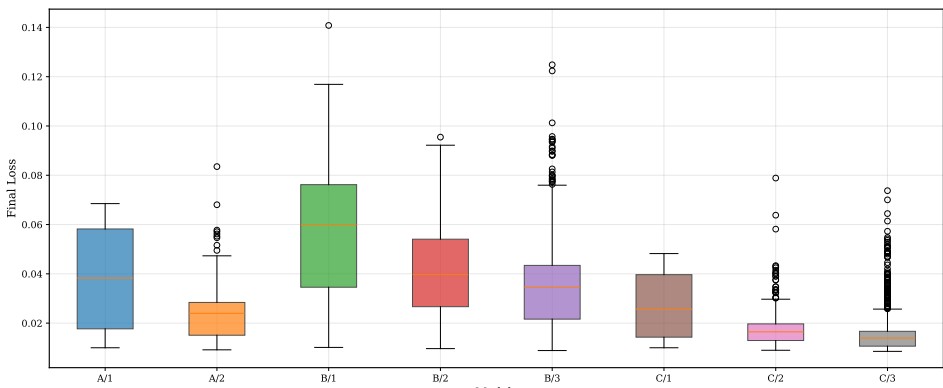

Figure 8: Model $X/E$ uses $X$ to denote the same model architectures and models used in Figure 4 and $E$ denotes the total number of experts (i.e. network sparsity). Increasing network sparsity decreases mean loss while increasing localized variance—especially as the number of experts reaches the input feature dimensions. This can attributed to the relatively unstable training of MoEs compared to dense models (despite training ten models for each cell and selecting the lowest loss).

The reconstruction for this theoretical MoE has zero loss only as the sparsity constraint holds (or goes to one in these toy models) because there is the chance more than $m$ features could be active at one time ($a \not\leq m$), which would exceed the monosemantic representational capacity of the network (but the dense polysemantic could do no better unless features are correlated in the distribution). Therefore, even if $a \not\leq m$ sometimes, the polysemantic model encounters the same problem and the monosemantic MoE under this construction may still outperform it under looser sparsity constraints.

Thus, for any dense model, $f_{\text{dense}}(x) = \text{ReLU}(Wx + b)$ under the sparsity constraint $|\text{support}(x)| \leq a$, there exists a MoE model $f_{MoE}(x)$ such that $f_{\text{dense}}(x) = f_{MoE}(x)$ for all valid inputs. In the toy settings described in this paper, the sparsity constraint holds in the limit where sparsity goes to one. However, in practice there may be an upper bound on the amount of features a particular amount of information can semantically encode, indicated by the size of meaningful embeddings of that data. Therefore, a MoE model with sufficient experts and a tractable amount of superposition (e.g. interpretable) may be sufficient to encode all features present.

### A.6 Limitations

Our results should be interpreted in light of several limitations. First, all experiments are conducted in controlled toy-model settings derived from superposition studies in sparse autoencoders (Elhage et al., 2022). While this enables precise measurement of interference and monosemanticity, it abstracts away many complexities of large-scale transformers, including multiple computation, attention, heterogeneous feature distributions, and realistic routing dynamics. Consequently, the transferability of our findings to real-world MoE architectures remains uncertain.

Second, our architectural and routing choices are intentionally restricted: we use simple experts, fixed top-$k$ routing (often $k = 1$), equal parameter budgets between dense and MoE models, and no auxiliary load-balancing losses. Practical MoEs often employ richer routing mechanisms, variable expert capacities, and multi-task objectives (Lepikhin et al., 2021; Fedus et al., 2022) or expanding beyond the task of reconstruction to next token prediction, for example, which may yield qualitatively different representational behaviors.

Third, the feature distributions in our synthetic tasks—including sparsity patterns, feature importance, and independence assumptions—are significantly simpler than those found in natural data.

Finally, although we observe increased monosemanticity and reduced superposition in MoEs under fixed conditions, we do not evaluate downstream performance trade-offs, training stability, or specialization dynamics at scale. Prior work suggests such dynamics can shift substantially with model size, optimization regime, and data diversity (Krajewski et al., 2024).

Overall, our study provides mechanistic insights under clean experimental conditions, but further work is required to validate these patterns in large, realistic MoE systems. Unfortunately, our methods do not scale naturally to larger models and this is a clear direction for future work in this space.

