# OpenReview forum: "Sparsity and Superposition in Mixture of Experts"
_ICLR.cc/2026/Conference — Submitted to ICLR 2026_

### Official Review · Reviewer_zaJm · 2025-10-28

**Soundness:** 1
**Presentation:** 1
**Contribution:** 2
**Rating:** 2
**Confidence:** 4

**Summary:**

This paper studies the superposition in toy MoE models. Most of its techniques are borrowed from a blog post by Elhage et al., which analyzes superposition in a toy dense model. Many toy setups, such as a hidden dimension of merely 1, are insufficient to support a full-length regular research paper.

The authors also observe that the phase change seen in dense models is absent in MoE models, and they study expert specialization and initialization. While the sections on specialization and initialization offer an interesting perspective, they still rely on a hidden dimension of 1. I appreciate the viewpoint here but remain unconvinced by the validity of all the results.

**Strengths:**

Section 5 presents an interesting perspective. If its findings could be validated in realistic MoE models, rather than the toy setups used, the paper would gain significant value.

**Weaknesses:**

1. The font in this paper’s template appears unusual and does not conform to the formal requirements of ICLR 2026.

2. This paper offers few novel techniques. Section 5 is interesting, but the rest of the paper merely adapts Elhage et al.’s method—originally applied to a dense toy model—to an MoE toy model, with formulations and definitions borrowed from Elhage et al. The evaluation metrics and analytical methods are also mostly identical.

3. It may be acceptable for Elhage et al. (whose work is merely a blog post) to use toy models with a hidden dimension of m=5. However, this paper uses m=6, or even fewer, m=1. Insights derived from such toy models are insufficient to serve as the core experiments of a regular paper.

4. In modern MoE architectures, SwiGLU is universally adopted as the expert structure, rather than the two-layer ReLU MLP used here. Activation functions and model architectures significantly influence model behavior and training dynamics, further rendering this paper overly "toy-like" in design.

5. On page 4, the conclusion that "The greater the number of experts, the less superposition in the model" is not informative. With more experts, the model is wider, and it naturally does not need to allocate features in a superposed manner. The related experiments are therefore uninformative to me.

6. In Line 220, the claim that the loss gap between the MoE and dense model in Figure 7 is "negligible" is incorrect. Every observed gap (on the order of 0.1) is significant.

7. The captions, labels, and discussions related to Figure 4 are confusing and hard to follow. Additionally, Figure 4A uses a setup with n=2, m=1; I cannot be convinced by experiments on models with a hidden dimension of 1.

**Questions:**

1. I do not understand why Figure 2b claims that MoE exhibits far less interference with other features than the dense model, as observed in Figure 1b. In both Figures 2a and 2b, a feature interferes with at most one other feature (there is only one blue dot per row or column).


2. To maintain the total parameter count, m=6 is split into three experts with m=2. Is it possible that the inherent separation of a dense MLP into MoE explicitly reduces superposition? In other words, can the reduced superposition in MoE be taken for granted?

3. While I consider Section 5 an interesting perspective, I really can not accept experiments using a hidden dimension of  m=1.

---

> ### Author Response · Authors · 2025-11-20
> **Response to Reviewer zaJm**
>
> We thank the reviewers for the detailed feedback and have updated the manuscript accordingly.
>
> **On Small Hidden Dimensions $m=6,1$**: We respectfully argue that superposition is a local phenomenon. In large models ($d_{model}=4096$), superposition does not happen 'globally' across all dimensions simultaneously; rather, it occurs within specific subspaces where features compete for capacity. Our toy models with $m=6$ are intended to isolate a single one of these subspaces to mathematically characterize the interference mechanics. Increasing $m$ to 4096 without changing the feature-to-dimension ratio would simply replicate these small-scale dynamics in parallel, making exact visualization ($W^\top W$) intractable without altering the fundamental physics of the interference.
>
> **On SwiGLU:** We acknowledge that SwiGLU is the standard for modern MoEs (e.g., Mixtral, Llama). However, the fundamental driver of superposition is the L2 reconstruction trade-off between storing a feature (reducing bias) and creating interference (increasing variance). This trade-off exists regardless of the specific non-linearity. Furthermore, SwiGLU is a gating mechanism that effectively increases local sparsity. Since our core finding is that network sparsity reduces superposition, we hypothesize that SwiGLU would likely amplify our results—providing the model yet another tool to 'eat the sparsity gap'—rather than negate them.
>
> **On Interference Differences:** While the dense model in Figures 1-2 ($m=6$) achieved relatively low interference (mostly pairwise), the key distinction is in **global** interference structure. In dense models (Figure 1b), zero interference between unconnected features results from optimization. In MoEs (Figure 2b), interference between features in different experts is **architecturally zero**. The MoE guarantees block-diagonal interference matrices, reducing potential interference scope from $N \times N$ to $\sum (N_e \times N_e)$—a significant reduction enabling easier monosemanticity.
>
> The architectural separation provides strong structural inductive bias against superposition. This is the fundamental mechanism improving interpretability: the architecture explicitly changes the optimization landscape to remove superposition incentives. By conditionally activating experts, the model aligns activation sparsity with feature sparsity, making dedicated monosemantic neurons optimal rather than compressed polysemantic ones.
>
> **Superposition**: Critically, our experiments maintain a constant total parameter budget (e.g., comparing a dense model with $m=20$ to an MoE with 20 experts of $m=1$), demonstrating that the observed reduction in superposition stems from the architecture's explicit partitioning of the feature space rather than an increase in model capacity.
>
> **Difference in loss:** We agree that the loss gap is non-zero and have updated the text to describe the performance as 'comparable' rather than 'negligible,' emphasizing that this slight increase in reconstruction error is the specific trade-off required to achieve the substantial gains in monosemanticity shown in our results.
>
> **Experimental Scale:** While Figure 5 used $m=1$, remaining experiments used $n \in [20, 100]$, $E \in [4, 10]$, and $m \in [5, 10]$. Table 1 aggregates results from one hundred trained models. We acknowledge that the first experiment (Figure 5) over-emphasized a correlation between input space distribution and model performance. We do not intend to argue for this at a large scale, but rather note an empirical observation on small models, in order to motivate the experiments that followed.
>
> **Figure 4 Clarification:** We sample $x_i \sim U(0,1)$ with sparsity $S \in [0,1)$ where $P(x_i = 0)= S$. To study phase transitions, we scale the last feature by $r \in \mathbb{R}^+$, so $I = (1, 1, ... r)$. Three setups with one active expert ($k=1$): (A) n=2, m=1; (B) n=3, m=1; (C) n=3, m=2. Using $m=1$ and $m=2$ isolates effects of interest: feature representation comparison across networks.
>
> Updated caption excerpt:
> > For a particular expert and input dimension (feature), we can decode how it is embedded in the hidden dimension—whether it is ignored (white), monosemantic (blue-purple), or superimposed (red). We plot joint feature norm ($||W_{n}||^2$) and superposition score ($\sum_{j < n} (\hat{W}_{n} \cdot W_j)^2$) across varying feature sparsity $S \in [0.1, 1]$ and relative last feature importance $r \in [0.1, 3]$, where the subscript ${n}$ denotes the last feature of $n$ total features.

---

> > ### Comment · Reviewer_zaJm · 2025-11-24
> >
> > Thank you very much for your response. While you have clearly explained the rationale for using small toy models, your discussion on the validation and generalization of these setups relies primarily on "we argue" and "we hypothesize" rather than experimental evidence. Since such argument alone does not fully convince me, I maintain my original ratings.

---

### Official Review · Reviewer_Qdj5 · 2025-10-28

**Soundness:** 1
**Presentation:** 2
**Contribution:** 1
**Rating:** 2
**Confidence:** 4

**Summary:**

The authors extend the mechanistic analysis of dense models by Elhage et al. to a toy mixture of experts setup. Through a newly introduced attempt to quantify monosemanticity, they show that large expert counts lead to more monosemantic features in their toy setup, and also that MoEs exhibit less of a phase transition as a function of sparsity. Ultimately, the authors advance our understanding of MoEs and tentatively advocate for their use as a more interpretable alternative to dense layers without performance degradation.

**Strengths:**

## [S1] Strong motivation + important problem

The paper addresses an important and timely issue in mechanistic interpretability. As the authors correctly note, the MoE architecture is quickly becoming the dominant paradigm for pushing capabilities, yet our understanding of how they work is in its infancy. This sets up the authors’ topic to be of high interest to both interpretability and capability researchers alike.

Unfortunately, as I discuss below, I see some issues with extending insights here to MoEs in practice, but the promise of the work’s premise still remains.

**Weaknesses:**

## [W1] Toy setup formulation needs polish and discussion about transfer to practice

The authors’ proposed toy MoE formulation for each expert as $\text{ReLU}({\mathbf{W}^e}^\top\mathbf{W}^e\mathbf{x}+b^e)$ is quite different to how MoEs are implemented in practice. Even compared to the original Sparse MoE [1], i see two important differences:

1. MoE expert’s input layers’ weights are **not tied to be symmetric**, like the authors’.
2. Each expert’s FFN often includes a second linear transformation after the ReLU [1].

The authors should discuss thoroughly how much the proposed symmetry constraint and omission of final linear transformation hinders our ability to extend insights to the non-toy settings? My concern here is that without explicit justification for why this indeed connects to practice, the authors’ insights might be heavily constrained to their unusual toy model formulation alone. Additionally, many SOTA MoEs in practice now use a shared expert [2,3,4]. The authors should comment on how superposition and/or their analysis is affected under this setup.

### Modifications needed

Importantly, the technical formulation of the toy MoE setup in Section 3 needs clarity and correction. This is necessary to make the authors’ setup perfectly legible to readers, given its non-standard nature. Whilst each issue alone may appear trivial, the presence of many such errors in presentation leads to the general impression that the paper lacks clarity, and precision--for a paper carefully studying a newly introduced toy setup, it is of paramount importance to clearly and correctly formulate the toy model they are proposing.

Some issues:

- [L111] this equation does not compute. $W_r^\top x$ would be needed (with the transpose) for this to work.
- On [L114], there is a confusing inconsistency between the use of the gating weights with $w_e$ and $w^e$ at once, which also clashes with the notation used for the input layer. The authors should define clearly how the normalization is computed, and I would suggest naming this something different entirely (e.g. $a_e$).
- $W_e$ is not defined, nor are its dimensions (used on [L112]).

## [W2] Missing related work section

The authors do not include a dedicated discussion of related work. Whilst 16 references do appear throughout the paper, a dedicated section is crucial to place the authors’ contributions in context of the prior literature.

As one example of why this is important, one of the authors’ key contributions is a definition of expert specialization for monosemantic features ([L066]). However, the authors do not discuss existing attempts to quantify expert monosemanticity in the literature, and why their proposed analysis offers additional insights; measured through ablations in [5,6]. A detailed discussion of how the proposed analysis relates to both existing works should be made to situate the work in relation to existing attempts.

---

## References

[1]: Shazeer, Noam M. et al. “Outrageously Large Neural Networks: The Sparsely-Gated Mixture-of-Experts Layer.” ICLR 2017

[2] Liu, A., Feng, B., Xue, B., Wang, B., Wu, B., Lu, C., ... & Piao, Y. (2024). Deepseek-v3 technical report. *arXiv preprint arXiv:2412.19437*.

[3] Team, K., Bai, Y., Bao, Y., Chen, G., Chen, J., Chen, N., ... & Zhang, H. (2025). Kimi k2: Open agentic intelligence. *arXiv preprint arXiv:2507.20534*.

[4] Meta AI. (2025, April 5). *The Llama 4 herd: The beginning of a new era of natively multimodal intelligence*. https://ai.meta.com/blog/llama-4-multimodal-intelligence/

[5]: Park, Jungwoo, et al. "Monet: Mixture of monosemantic experts for transformers." ICLR 2025.

[6]: Oldfield, James, et al. "Multilinear mixture of experts: Scalable expert specialization through factorization." NeurIPS 2024.

**Questions:**

## [Q1] Mixed definitions of monosemanticity / features

On [L086], the authors state `Monosemantic features are defined as those that are well-aligned with individual neurons`.

I am a little confused by this definition. In Elhage, monosemanticity is a property of *neurons* (possibly SAE latents), not the high-level concepts; the goal is to establish the independent computational units of meaning. The difference between the two appears to me important.

For example, there may exist multiple neurons that monosemantically correspond to the *same* high-level concept. This is consistent with the Elhage definition of monosemanticity, but not the authors’, when formulated as a property of the concept.

Furthermore, “features” is used on [L086] to refer to human-interpretable concepts, but again on [L110] onwards to denote the $n$ input neurons. Monosemanticity and superposition suggests that this equivalence does not hold.

Might the authors please clarify their use of the terminology here?

## [Q2] Load balancing mixed use

The authors should comment on why load balancing is used for Sect. 4 but not for Sect. 3. At the minute, it is left unexplained; and the extent to which experts are balanced should surely influence the kinds of features it learns.

Specifically, without a load balancing loss for the experiments in Sect. 3, what is preventing the MoE from learning a single expert alone (functionally equivalent to the dense model)? Might the authors please comment on the balance observed in the first section?

**Details Of Ethics Concerns:**

None of note.

---

> ### Author Response · Authors · 2025-11-20
> **Response to Reviewer Qdj5**
>
> We thank the reviewers for the detailed feedback on the paper and have updated the manuscript accordingly.
> - We have clarified our terminology in the background section to distinguish between monosemantic *neurons* (as they are traditionally defined in Elhage et. al) and monosemantic *features* (which, for brevity, we use to describe features with less superposition). The relevant paragraph from the background:
>
> > Monosemanticity is a characteristic of individual neurons, where a neuron's activation cleanly corresponds with a single $\alpha_i$ (i.e., features are basis-aligned). When features are orthogonal (not in superposition) but not basis-aligned, neurons remain *polysemantic* even though feature interference is minimal. In this paper, we focus on reducing superposition rather than enforcing basis-alignment. For brevity, when we describe models, experts, or features as ``more monosemantic," we mean they display less superposition.
> >
>
> - The decision to omit load balancing in Section 3 but include it in Section 4 was deliberate, driven by the differing goals of each experiment.
> - The goal of Section 3 is to characterize the *intrinsic* inductive bias of the MoE architecture regarding superposition. We wanted to measure whether experts specialize and reduce interference *naturally*, without being forced to do so by an auxiliary loss. Using a load-balancing loss here would confound the results, as it artificially forces the model to distribute features, potentially masking the architecture's architectural bias (or lack thereof) toward monosemanticity.
> - In the regime of Section 3 (uniform feature density, many features), we found that standard random initialization is sufficient to split features across experts. Because a single collapsed expert would suffer from high interference (high reconstruction loss), there is a strong gradient incentive for unused experts to capture subsets of features where they can offer lower interference than the dominant expert. We monitored expert usage in these experiments and observed that while not perfectly uniform, experts remained active and utilized without auxiliary loss.
> - Section 4 involves sweeping across extreme regimes of sparsity and feature importance (e.g., very high sparsity or single dominating features). These edge cases are highly susceptible to expert collapse, when experts and routers fall into local minima where certain experts are entirely ignored. To generate a complete and stable phase diagram, the auxiliary loss was necessary to ensure convergence across all hyperparameters.
>
> - We agree that the notation required standardization and have overhauled Section 3.1 to ensure mathematical precision.
> - 1. Toy Formulation vs. Practice (Tied Weights & ReLU position): Our experts use tied weights ($W_{dec} = W_{enc}^\top$) and place the ReLU at the output, whereas production MoEs use untied weights and placing the non-linearity between layers ($W_2(\sigma(W_1 x))$). Our formulation strictly follows Elhage et. al’s setup. The motivation for tied weights is to isolate the geometry of feature packing from the rotational degrees of freedom introduced by untied matrices. Similarly, placing the ReLU at the output is mathematically necessary in this specific toy setting to allow the model to “clean up” interference noise (negative dot products) arising from superposition. While real MoEs differ architecturally, the fundamental constraint driving superposition is the **bottleneck dimension (**$m$**) vs. feature count (**$n$**)**. Whether the weights are tied or not, this capacity constraint forces compression. Therefore, we argue that the *incentives* for monosemanticity (”eating the sparsity gap”) derived here remain applicable to untied setups, even if the resulting geometry is harder to visualize.
> - 2. Shared Experts: Regarding shared experts, we hypothesize that their inclusion would likely amplify our findings. A shared expert effectively handles the “dense” (commonly occurring) features. By offloading these polysemantic/dense features to the shared expert, the routed experts are left with an even sparser residue of features, which should theoretically drive them toward even higher monosemanticity.
> - 3. Notation Corrections: We have corrected the inconsistencies in Section 3.1. We now explicitly define $x$ as a column vector, $W_r \in \mathbb{R}^{E \times n}$ and gating weights as $w_e$.
>
> - Due to space constraints, the related work section is not reproduced here but will be added to the new manuscript.

---

> > ### Comment · Reviewer_Qdj5 · 2025-11-23
> >
> > Thanks to the authors for their thorough response! Some comments:
> >
> > 1. Thanks for making the usage of terms more clear, I think this has helped with the clarity of the manuscript.
> >
> > 2. The authors write:
> >
> > > We wanted to measure whether experts specialize and reduce interference *naturally*, without being forced to do so by an auxiliary loss
> >
> >   The authors emphasize the term “naturally” here, yet I am unsure what their intended message is with this word -- might they please elaborate? For example, one could argue a “natural” setup is one *with* a load-balancing loss (as almost all MoEs currently use one in practice).
> >
> > More generally, I don’t believe the inclusion/exclusion of a load-balancing term to be critical, so long as a justification like the one the authors presented is made in a revised manuscript.
> >
> > 3. The authors state that `Our formulation strictly follows Elhage et. al’s setup.` , but I do not think this is a convincing response unfortunately -- why is following the Elhage setup (leading to a non-standard MoE implementation) more insightful than adopting a popular MoE model and studying to what extent sparsity and superposition interact here, in practice? Rather than relying on arguments about similar incentives, experimental results in a real MoE setup would be much more convincing in my opinion.
> > 4. Thanks to the authors’ thoughts on shared experts; their response sounds reasonable, and I think it suffices to note this in the limitations section to scope the study.
> > 5. The authors have not yet added the “related work” section they allude to, so I cannot comment on this. However, I must note that the current revised manuscript is almost 11 pages (beyond the permitted 10 page rebuttal/discussion limit).
> >
> > Thanks again to the authors for their willingness to reply to the reviews and improve the paper in light of the critical feedback; I think it is an interesting and important topic and area to study, and believe it will eventually form a useful and impactful contribution (with sufficient work to address the reviews)!

---

### Official Review · Reviewer_JRyu · 2025-10-28

**Soundness:** 2
**Presentation:** 1
**Contribution:** 2
**Rating:** 2
**Confidence:** 3

**Summary:**

This paper applies the superposition framework of Elhage et al to MoEs, in the context of toy autoencoders. The main claims are: (1) MoEs represent the same number of features as do dense models with the same capacity, but they do so more monosemantically, meaning with less superposition or interference between features. (2) MoEs don't show the same phase transitions that dense models do between monosemantic representation, polysemantic representation and ignoring features, as a function of feature sparsity in the input distribution and imbalance in the feature weights on the loss function.

**Strengths:**

The approach potentially gives insight into representation and expert specialization in MoEs

**Weaknesses:**

I had great difficulty figuring out what was done in many parts of the paper. I don’t normally share such detailed notes as I do in the Questions section, but in this case I do to help explain how much work this paper needs.

The main claim that “MoEs represent the same number of features as the dense model, but more monosemantically” (e.g., L220) seems impossible. How can two models match in the number of features they represent and the number of dimensions (“parameters”) they use, but differ in the number of features per dimension (i.e., in monosemanticity)?

Mathematically, the definitions of feature representation (L141) and features per dimension (L204) are nearly the same. If we stack $W=W^{1:E}$ we get an $Em\times n$ matrix matching the dense model. Setting aside marginal load imbalance (i.e., assuming $p_e=k/E$) the definition of features per dimension is the same for the two models (i.e., stacked and unstacked representations). Moreover the result equals the sum of squared feature strengths: $|W|_F^2 = \sum_i |W_i|^2$ for the dense model and $|W|_F^2 = \sum_i \sum_e |W^e_i|^2$ for the MoE. So if the models really match on summed feature strengths and differ on features per dimension, it seems like this can only be because of the different orders of summation and squaring in how the measures are applied to the two models (and this should be spelled out in the paper), but that would hinge on the seemingly arbitrary choice to define feature strength as $|W_i|$ rather than $|W_i|^2$. Also the fact that strengths in figs 1a and 2a are all near {0,1} suggests that squaring makes little difference.

Putting all this somewhat differently, if superposition happens when a network encodes more features than it has hidden dims then MoE (with the same total number of hidden dims as the dense model) can’t help: monosemanticity will still require dropping some features.

**Questions:**

L58: What phase change, i.e. what is the macroscopic variable and what are the hyperparameters?

L63: It looks like the hyperparameter for the phase change is network sparsity, but that doesn’t apply to dense models.

L87: Does it have to be a single neuron, or can it be an oblique direction in activation space? It should be the latter because of rotation invariance (for standard FF layers). The important criterion is that features have orthogonal representations.

L102: What claim? The previous sentence is just descriptive. I also don’t understand the second half of the sentence: do you want to quantify representational similarity between experts? (I don’t think that’s the topic of this paper.)

L111: $n$ is the number of input features, not the input features (those are denoted $x$)

L112: $W_r$ is $E\times n$ not $n \times E$. It would also help to state $W^e\in\mathbb{R}^{m\times n}$.

L114: $w_e$, $w^e$

L116: What is the loss? From what comes next I think it’s squared error with weights $I$. Also what is the dataset or generating distribution? What is the optimization/training procedure? These are critical questions for understanding all the experiments in the paper.

L140: $W^e_i$ is column $i$ of $W^e$?

There seems to be an assumption that $|W^e_i| \le 1$. I can see this for $E=1$ because otherwise the reconstruction overshoots. But in MoE the reconstruction is weighted by the gating weights which are $<1$ (see def of $x’$ at L114). For example if a feature is represented by only one expert $e$ then that expert would need to scale up its output by $1/w_e$.

Does the term ‘dense model’ mean anything more than $E=1$?

How is superposition in figs 1a and 2a defined?

L146: What does “roughly the same number of features” refer to? I see 10 features (0,1,3,5,6,7,8,9,10,12) represented in 1a and 8 features (0,1,2,3,4,5,6,9) in 2a.

In what sense do the models have equal total parameters? They have an equal number of hidden dimensions (6) but the MoE also has gating weights.

Figs 1b 2b are described as measuring interference but they don’t match the definition at L143. Also the claim is less interference in the MoE but I count 4 interfering pairs in fig 1b and 5 in 2b.

L204 (please consider numbering equations): This expression doesn’t work as a count if $|W^e_i| > 1$ (see comment above) because then feature $i$ contributes more than 1 to the count.

L262: I think you mean $x=(x_1,x_2,\dots,rx_n)$. Also it’s poor notation to define the $n$th component of $x$ as $rx_n$.

L263: $x_i\sim U(0,1)$

L263: "$S$ likelihood that $x_i=0$" doesn’t make sense. You want to define a mixture distribution between a uniform and a point mass.

L267: This should be stated formally especially given earlier confusion about param counts. I think you mean $m_{\rm dense} = km_{\rm MoE}$.

Equating active dimensions ($m_{\rm dense} = km_{\rm MoE}$) doesn’t seem like the right comparison (as opposed to $m_{\rm dense} = Em_{\rm MoE}$) because it gives the MoE more capacity than the dense model. The MoE can represent a feature with some expert and choose not to activate that expert when that feature is absent. So it’s not clear whether the differences regarding phase transition in fig 4 are due to dense vs MoE per se or due to differences in model capacity.

Fig 4: I think the network diagrams for ABC are meant to indicate the values of $n$ and $m$. The implication about $m$ is ambiguous so it would be better just to state the values. Also what are the values of $k$? (Ok we are eventually told $k=1$.)

Each pixel in fig 4 is a completely separate simulation, and each model should be invariant to permutation of the experts, so how can there be systematic differences between experts 1 and 2 in the first two columns?

Why is the importance of only one feature being varied? I suspect the figure is showing values only for that feature but the caption suggests otherwise (subscript $i$ instead of $n$).

L368: Does “feature $x$” indicate $x$ denotes a unit vector $e_i$ (i.e., $x_j = \delta_{ij}$) or does “feature” here refer to any input vector (i.e., arbitrary $x\in\mathbb{R}^n$ or perhaps $x\in[0,1]^n$)?

L371: What measure is used to define volume? Induced Lebesgue measure on the L2 unit sphere?

L373: This claim (“they tend to align experts with particular features”) is not warranted by the tiny sample shown in fig 5. It requires a systematic study, also using more feature dimensions since with $n=2$ it’s nearly impossible to have good load balancing without allocating $x=(1,0)$ and $x=(0,1)$ to different experts.

L409: I don’t think $c_i$ has been defined. Is this a statement about $W_r$ from L111? More importantly, how can the gate matrix be the diagonal (I assume you mean an identity matrix) when it isn’t square ($W_r\in\mathbb{R}^{E\times n}=\mathbb{R}^{5\times20}$)?

Fig 5: what do the colors represent? (Probably separate questions for even and odd columns)

---

> ### Author Response · Authors · 2025-11-20
> **Reviewer JRyu (part 1)**
>
> We thank the reviewer for the detailed feedback on the paper.
> - **L220, features per dimension & parameters:** The ratio of number of features represented to total parameters is indeed constant across both models. However, monosemanticity is determined by feature interference. In a dense model, every feature potentially interferes with every other feature, creating global pressure to use superposition (non-orthogonal packing) to resolve conflicts. In a MoE, the router effectively partitions the feature set into disjoint subsets assigned to specific experts. Consequently, a feature in Expert A only competes for representational capacity with the small subset of features also routed to Expert A, not the entire global set. This reduced *local* interference allows for monosemanticity rather than resorting to superposition, even if the global parameter budget is identical.
> - As stated by Elhage et. al, superposition exploits the gap between the high sparsity of features (which are rarely active) and the density of standard neurons (which fire or are processed on every forward pass). Dense models must use superposition to pack sparse features into limited active dimensions.
> - MoE architectures, however, 'eat this same gap' explicitly through conditional computation. By organizing neurons into blocks that only activate a fraction of the time, MoEs align the model's activation sparsity with the underlying feature sparsity. If a model only expends compute on active features, splitting polysemantic neurons into dedicated monosemantic neurons becomes the optimal strategy. By approximating this regime, our MoEs relieve the optimization pressure to superimpose, naturally converging to monosemantic representations even when the global parameter budget is identical to the dense baseline.
> - **Frobenius norm & Feature norm equivalence:** Under the assumption of balanced loading, the features per dimension metric is algebraically proportional to the total squared feature strength per total model dimension. Consequently, the difference in this metric observed in Figure 3 reflects a real difference in the total feature strength learned by the models. Specifically, the dense model utilizes superposition to learn a number of features significantly exceeding the number of dimensions (increasing the numerator). In contrast, the MoE models tend to converge to a monosemantic regime where the number of learned features is capped by the number of experts ($n_{learned} \approx E \cdot m$), resulting in a lower total feature strength but zero interference.
> - This interpretation explains why the dense model achieves slightly lower loss in Figure 6: it is indeed representing more features, whereas the MoE trades off representational quantity (ignoring excess features) to maximize monosemanticity for the features it does learn. We have revised the manuscript to explicitly distinguish between the capacity-constrained regime (Fig 3) and the sufficient-capacity regime (Fig 2), clarifying that MoEs prioritize monosemanticity over raw feature count.
> - $W_i^{e}$ refers to the i-th column of the encoder matrix $W^e$, representing direction of feature $i$ in the expert’s hidden space.
> - **feature norms magnitude:** The gating weights are renormalized over the selected experts. For $k=1$, the selected expert has a renormalized weight of $w_e = 1$ which makes $\|W_i^{e}\| \approx 1$ sufficient. Even for $k>1$ $(\implies w_e <1)$, increasing the norm $\|W_i^{e}\|$ comes at the cost of amplified interference from features in superposition. Scaling the norm of the feature weight vector also corresponds to higher dot products with other features, $(W_i^e \cdot W_j^e)$. Empirically, for $k>1$, we find that feature norms rarely exceed 1 as shown in fig 2 (a).
> - We acknowledge that equation 2 (features per dim) relies on the assumption that learned features have approximately unit norm, i.e., $\lVert W_i^e \rVert \approx 1$. Mathematically, if feature norms significantly exceeded 1, this expression would indeed overestimate the feature count. However, in the context of these toy models, we find that the optimization landscape strongly discourages norms significantly larger than 1 due to the trade-off between signal recovery and interference. As the reviewer notes, one might hypothesize that for $k>1$ (where gating weights $w_e < 1$), the expert might scale up $\lVert W_i^e \rVert$ to compensate. However, increasing the norm of a feature vector simultaneously scales up its dot products with all other features, $\langle W_i^e, W_j^e \rangle$, amplifying interference noise. Empirically, we find that the model prefers to keep norms bounded to minimize this interference. As shown in Figure 2(a), feature norms for learned features cluster tightly around 1.0, even in $k>1$ settings where renormalization handles the signal magnitude. Thus, under these empirical conditions, the Frobenius norm squared serves as an accurate proxy for the count of learned features.

---

> ### Author Response · Authors · 2025-11-20
> **Response to Reviewer JRyu (part 2)**
>
> - We define “dense” model as simply the specific case in our MoE where $E=1$.
>
> - The superposition color is given by $\sum_{j} (\hat{W}_i \cdot W_j)^2$ in fig 1(a) and 2(a)
>
> - **Figure 5:** We acknowledge that section 5 over-emphasized a correlation between input space distribution and model performance. This section is not intended to argue for this at a large scale, but rather note an empirical observation on small models, in order to motivate the experiments that followed. With 3 total experts ($E=3$) in this initial experiment, each expert is mapped to a separate color. An individual pixel represents an input vector $x \in R^2$, and it’s color represents the expert this vector is routed to (since $k=1$). See the updated relevant paragraph:
>
> > In small models ($n=2, m=1, E>1$), we empirically find that models that distribute the input space across more experts tend to achieve lower loss, testing with $E \in [2, 7]$. Holding $n=2$ allows us to visualize which portions of the input space get routed to which experts, as seen in Figure 5. This warrants a question: does the allocation of the input space to certain experts imply any characteristics regarding those experts? Our definition of expert specialization suggests that this allocation implies *monosemanticity*, which we will see is a correlation that holds for larger toy models (e.g, m=10).
> >
>
>
> - **Figure 6:** We used the term “diagonal” to refer to the main diagonal being filled with 1’s; in the case of a square matrix, this would be the identity matrix. However, in Figure 6, this refers to the first $E=5$ columns being the basis vectors. This has been clarified in the revised manuscript. $c_i$ was used improperly and has been replaced with $W^r_i$
>
> We thank the reviewer for their close inspection of the visualizations.
> 1. Feature Counts: We agree that in this specific run, the Dense model represents 10 features while the MoE represents 8. We used the phrase "roughly the same" to indicate they are of comparable magnitude (both $\approx 1.5 \times$ the dimension count), but we have revised the text to be precise about the trade-off: the MoE represents slightly fewer features but isolates them more effectively.
> 2. Parameter Equality: The "equal parameters" claim refers to the representational capacity of the hidden layers ($m_{\text{dense}} = \sum m_{\text{experts}} = 6$). We acknowledge that the MoE includes additional parameters for the gating network ($W^r \in \mathbb{R}^{E \times n}$). We have clarified in the text that we match the total hidden capacity.
> 3. Visualizing Interference: You are correct that Figures 1b/2b show the Gram matrix $W^\top W$, which reveals the pairwise cosine similarities. We have updated the text to clearly distinguish the visualization (Gram matrix) from the metric (sum of squared off-diagonal elements).
> 4. Counting Interference: While the MoE in Figure 2b shows local interference within experts, the global interference structure is fundamentally different. In the Dense model, any feature can interfere with any other feature. In the MoE, interference is strictly block-diagonal; a feature in Expert 0 has zero interference with features in Expert 1 or 2. We have refined our claim to emphasize that MoEs reduce the global scope of interference via partitioning, even if local superposition persists within experts.
>
> The phase change section, especially the experimental setup, notation, and caption, have been overhauled to improve clarity and mathematical precision.
>
> - The phase change is in the representation of the last (*n*th)feature, measured jointly across feature norm ($||W_i||2$) and superposition score $\sum_{j < n} (\hat{W}_{n} \cdot W_j)^2$, and the parameters are overall feature sparsity and last feature relative importance. We then compare phase change plots across different network sparsity and architecture (where dense models have 0 network sparsity).
> - We take $x_i \sim U(0,1)$, except for a given sparsity $S \in [0,1)$, $P(x_i = 0)= S$. To isolate and study phase transitions for a single feature, we scale the magnitude of the last feature by a factor $r \in \mathbb{R}^+$, such that $I = (1, 1, ... r)$. This allows us to test expert sensitivity to feature magnitude.
> - There are three models setups, all with one active expert ($k=1$): (A) n=2, m=1; (B) n=3, m=1; and (C) n=3, m=2.
> - We fix active parameters ($m_{dense}=km_{moe}$, ignoring router parameters) to compare within model architectures; otherwise, any observed differences could be attributed to architectural changes instead of the number of active parameters. Coincidentally, 4.C.1/1 has the same number of active parameters as 4.B.X/2. But the latter has only one hidden dimension ($m=1$) to encode the same number of input features ($n=3$) as the former, with two hidden dimensions ($m=2$), making it difficult to use superposition to understand specialization.

---

> ### Author Response · Authors · 2025-11-21
> **Response to Reviewer JRyu (part 3)**
>
> - In Section4, we vary only one feature for clarity, compute, and because another effectively just makes a more complex relative importance variable—it doesn’t help us determine how representations are embedded as a function of network sparsity. (Notation has been corrected for $x_i$ to $x_n$, the last feature.)
>
> Clarifying why we would expect each model should be invariant to permutation of the experts:
>
> - The experts are architecturally symmetric and invariant to permutation. However, we hypothesize the consistency observed in Figure 4 is a result of the continuity of the optimization landscape under fixed initialization seeds. While the experts are structurally identical, their random initializations differ. The regions of consistent color in the phase diagram indicate basins of attraction: for a specific range of sparsity and importance, the specific initialization of expert 1 (dictated by the seed) consistently positions it more favorably to learn the feature than expert 2.
>
> We have revised the manuscript to include all of the above changes. Thank you for the detailed questions and feedback on the paper.

---

> > ### Comment · Reviewer_JRyu · 2025-11-24
> >
> > Thank you for the detailed replies and revisions to the manuscript.
> >
> > I'm confused by the first 3 points in the reply. "In a dense model, every feature potentially interferes with every other feature, creating global pressure to use superposition (non-orthogonal packing) to resolve conflicts" — Isn't this backwards? Interference happens precisely when feature encodings are non-orthogonal. The original Elhage et al point was that such interference is not too harmful when features are sparse. "If a model only expends compute on active features, splitting polysemantic neurons into dedicated monosemantic neurons becomes the optimal strategy" — This would make sense if the dense and MoE models were matched in active dimensions ($m$). That is, if every expert has the capacity of the full dense model (and the features are sparse) then the MoE has the luxury of representing every feature monosematically. But this is not very interesting. In the more relevant case where the models are matched on total dimensions ($Em$), the quoted statement doesn't make sense.
> >
> > Points 4-5 in the reply make much more sense and go against points 1-3 (to the extent I understand what you're arguing with points 1-3). To rephrase points 4-5, MoEs don't take advantage of superposition and instead simply ignore some features, and this hurts their performance. So the MoE is more monosemantic, but not in a useful way. I can't tell whether you agree with this conclusion, but the paper's main message seems to be different, instead claiming that the MoE architecture encourages monosemanticity and hence yields greater interpretability without sacrificing performance.
> >
> > The "counting interference" point in part 2 of the reply seems to reveal a similar confusion as above. The block-diagonal structure of the interference matrix is not an extra affordance of the MoE architecture; it's a constraint. Embeddings $W^e\_i$ and $W^{e'}\_j$ (for $e'\ne e$) are forced to be orthogonal, whereas in a dense model $W_i$ and $W_j$ could be orthogonal or not. If the dense model happened to partition its dimensions such that each feature was represented in just one part of the partition, then its interference matrix would also be block-diagonal. However, the dense model doesn't have to do this. Instead the model can take advantage of superposition across all its dimensions.
> >
> > If we're in agreement that the main conclusion is that MoEs fail to take advantage of superposition, then it would be interesting to ask why. One conjecture is that it's harder to pack extra features into lower-dimensional spaces. For a concrete example, a dense model with 20 dimensions might be able to represent 25 features with little interference while an MoE with 20 dimensions spread over 4 experts would need to pack 5 features into each 4d space. The "packing ratio" is 125% for both models but a fixed ratio is probably harder to manage with a smaller space, as suggested by the Johnson–Lindenstrauss lemma which shows the number of near-orthogonal vectors grows exponentially (not linearly) with the vector space dimension.

---

### Official Review · Reviewer_SA9i · 2025-10-30

**Soundness:** 2
**Presentation:** 2
**Contribution:** 2
**Rating:** 2
**Confidence:** 3

**Summary:**

This paper extends the ideas presented in the anthropic blog, with a focus on toy MoE models in the context of superposition.
The authors examine the superposition of these toy MoE models and observe the absence of a phase transition, which is seen in dense models.
Additionally, the paper explores expert specialization and initialization, which provides insights to the understanding of MoE behavior.

**Strengths:**

- The exploration of expert specialization and initialization is interesting to me.
These topics provide insights into a better understanding of MoE behaviors.

- The authors conduct extensive experiments to support their idea.

**Weaknesses:**

- Some of the paper is a straightforward extension of the anthropic blog, which adapts the research on dense models to MoE models.
As a result, the contribution feels somewhat limited.

- The authors' findings on toy models are interesting, but they are not entirely convincing to me due to the experimental setups.
Firstly, the experiments are conducted on toy models with a very small hidden dimension (e.g., 6 or even 1).
While interesting, it is hard for me to trust the conclusions draw from such a toy models;
Moreover, it is unclear whether the conclusions drawn from the two layer MLP with ReLU are truly relevant to modern FFNs, such as SwiGLU, which limits their applicability to real-world MoEs.

**Questions:**

---
Q1: Would it be possible for the authors to conduct the experiments in Figure 5 with different configurations?

The current results, set m=1, do not provide sufficient evidence to convince me.

---

Q2: I find Figure 4 to be somewhat complex and difficult to interpret.
Could the authors consider revising the caption, improving the labels, and providing additional clarification on the experimental design.

---

> ### Author Response · Authors · 2025-11-20
> **Response to Reviewer SA9i**
>
> We thank the reviewer for the detailed feedback.
> - While our work builds on Elhage et. al., the scientific value lies precisely where the MoE architecture **deviates** from the dense baseline established by Elhage et al.:
> - **Disappearance of Phase Changes:** We demonstrate that MoEs do not exhibit the discrete phase changes characteristic of dense models. This is a non-trivial finding: it suggests that network sparsity fundamentally alters the optimization landscape from a 'discrete choice' regime to a continuous one.
> - **Competing Mechanisms for Sparsity:** We provide a theoretical unification of MoEs and superposition, arguing that they are competing solutions for exploiting feature sparsity (”eating the same gap''). This explains *why* MoEs are naturally more monosemantic—a theoretical insight absent in previous work.
> - **Redefining Specialization:** The literature typically defines expert specialization via token-level load balancing. We propose and validate a mechanistic definition based on **monosemantic feature geometric occupancy**, showing that proper initialization can drive specialization without auxiliary loss.
> - Given the dominance of MoEs in current SOTA LLMs, verifying whether mechanistic principles derived from dense models hold in sparse architectures is a critical step for the field.
> - We thank the reviewer for raising the gap between toy models and modern large-scale architectures.
> - 1. On Small Hidden Dimensions ($m=6$): We respectfully argue that superposition is a local phenomenon (Scherlis et.al, 2025 in References). In large models ($d_{model}=4096$), superposition does not happen 'globally' across all dimensions simultaneously; rather, it occurs within specific subspaces where features compete for capacity. Our toy models with $m=6$ are intended to isolate a single one of these subspaces to mathematically characterize the interference mechanics. Increasing $m$ to 4096 without changing the feature-to-dimension ratio would simply replicate these small-scale dynamics in parallel, making exact visualization ($W^\top W$) intractable.
> - 2. On ReLU vs. SwiGLU: We acknowledge that SwiGLU is the standard for modern MoEs (e.g., Mixtral, Llama). However, the fundamental driver of superposition is the L2 reconstruction trade-off between storing a feature (reducing bias) and creating interference (increasing variance). This trade-off exists regardless of the specific non-linearity. Furthermore, SwiGLU is a gating mechanism that effectively increases local sparsity. Since our core finding is that network sparsity reduces superposition, we hypothesize that SwiGLU would likely amplify our results—providing the model yet another tool to 'eat the sparsity gap'—rather than negate them.
>
> We acknowledge that section 5 over-emphasized a correlation between input space distribution and model performance. This section is not intended to argue for this at a large scale, but rather note an empirical observation on small models, in order to motivate the experiments that follow. See the updated relevant paragraph:
>
> > In small models ($n=2, m=1, E>1$), we empirically find that models that distribute the input space across more experts tend to achieve lower loss, testing with $E \in [2, 7]$. Holding $n=2$ allows us to visualize which portions of the input space get routed to which experts, as seen in Figure 5. This warrants a question: does the allocation of the input space to certain experts imply any characteristics regarding those experts? Our definition of expert specialization suggests that this allocation implies monosemanticity, which we will see is a correlation that holds for larger toy models (e.g, m=10).
> >
>
> For section 4, we have updated the caption, improved the clarity overall, and included more specificity in the experimental setup—all of which should also clarify the labels.
>
> - We take $x_i \sim U(0,1)$, except for a given sparsity $S \in [0,1)$, $P(x_i = 0)= S$. To isolate and study phase transitions for a single feature, we scale the magnitude of the last feature by a factor $r \in \mathbb{R}^+$, such that $I = (1, 1, ... r)$. This allows us to test expert sensitivity to feature magnitude.
> - There are three models setups, all with one active expert ($k=1$): (A) n=2, m=1; (B) n=3, m=1; and (C) n=3, m=2.
>
> Part of our improved Figure 4 caption, where we use the subscript n to denote we are interested in the last feature, the only feature where we change relative importance:
>
> > For a particular expert and input dimension (feature), we can decode how it is embedded in the hidden dimension—whether it is ignored (white), monosemantic (blue-purple), or superimposed (red). We plot joint feature norm ($||W_{n}||^2$) and superposition score ($\sum_{j < n} (\hat{W}_{n} \cdot W_j)^2$) across varying feature sparsity $S \in [0.1, 1]$ and relative last feature importance $r \in [0.1, 3]$, where the subscript ${n}$ denotes the last feature of $n$ total features.

---

### Official Review · Reviewer_z1ER · 2025-11-01

**Soundness:** 2
**Presentation:** 1
**Contribution:** 2
**Rating:** 2
**Confidence:** 4

**Summary:**

This paper studies how mixtures of experts learn some toy models of data, versus how dense MLPs learn them. The paper finds that mixture of experts neurons are much more monosemantic than dense MLP neurons.

**Strengths:**

* The question of how the mixture of experts architecture interacts with how concepts are represented by neurons (monosemantically vs. polysemantically) is interesting.

* The analyses of this paper look like they are probably quite interesting. I just had a very tough time reading them because the basic definitions and setup are not presented.

**Weaknesses:**

* Some definitions are missing, which makes the paper unclear in cases and hard to read in others. (See my questions below for examples.) The paper could greatly benefit from a clearer exposition of definitions so that readers can understand what the authors concretely mean by a "feature", or by "monosemanticity" in this context.

* The analyses are all conducted on toy models, without any analysis, e.g. of MoE models trained on real data.

**Questions:**

* What does importance I = 0.7^i mean? What is the data distribution in the experiments? This is not described until much later on in the lines 260-264, but I am not sure how to parse this definition. Are the features vectors? Why are they scalars in this definition? Why is only the last feature sampled from a different range from the other ones?

* What do the colors of the bars mean in Figure 1(a), Figure 2(a)? Are these D_i scores from 0 to 1?

* The architecture choice in section 3.1 doesn't make sense to me. Why is the ReLU applied outside of everything else when computing x_e' ? Doesn't that mean that the architecture will always output reconstructions with nonnegative entries?

* How is a feature defined as monosemantic or not?

---

> ### Author Response · Authors · 2025-11-20
> **Response to Reviewer z1ER**
>
> We thank the reviewer for pointing out the ambiguity in our definitions. We have revised our precise experimental setup in Section 3.1 which are described below:
> - 1. Meaning of $I = 0.7^i$: This notation describes an exponential decay in feature importance. Specifically, the $i$-th feature in the input vector is assigned a weight $I_i = 0.7^i$ in the loss function. This creates a hierarchy of utility: the model reduces loss significantly more by reconstructing feature 0 than feature 20. This forces the model to prioritize which features to store when capacity is limited.
> - 2. Vectors vs. Scalars: The input $x \in \mathbb{R}^n$ is a high-dimensional vector. We refer to the individual components $x_i$ (scalars) as "features" because, in this idealized setup, the input basis is disentangled. However, the learned representation of feature $i$ is a vector $W_i \in \mathbb{R}^m$ (the $i$-th column of the encoder).
> - 3. The Last Feature importance: This applies only to the specific "Phase Change" experiments (Section 4), where we vary the magnitude $r$ of the last feature importance to trace the precise boundary between learning and ignoring a feature. For the general superposition experiments (Section 3), features are sampled identically from $U(0,1)$ conditional on some sparsity and relative importance.
> - The color bars in Fig 1 (a) and 2 (a) represent the **superposition (interference) score**, quantified by the sum of squared projections of all *other* features onto the feature in question: $\sum_{j \neq i} (\hat{W}_i \cdot W_j)^2$.
> They are **not** the $D_i$ scores, but they are inversely related.
> - $D_i$ **(Dimensionality)** measures the fractional dimension dedicated to a feature; a monosemantic feature has $D_i \approx 1$.
> - **The Color (Interference)** measures how much other features intrude on this feature's direction; a monosemantic feature has an interference score of $\approx 0$ (indicated by purple/dark blue in the figures). Features with high interference (green/yellow) have low $D_i$ scores.
> - The architectural choice of ReLU constrains the reconstruction to be non-negative. This is intentional for two reasons:
> - 1. **Data Constraint:** Our synthetic input features are sampled from a distribution where $x_i \sim U(0,1)$ (with sparsity). Since the ground truth features are strictly non-negative, a valid reconstruction must also be non-negative.
> - 2. **Mechanism of Superposition:** Following the Elhage et.al’s setup, placing the ReLU at the output is the critical mechanism that enables the model to hide interference. When features are packed non-orthogonally, they create 'interference noise' (nonzero dot products). The model learns to orient features such that this interference is often negative (e.g., antipodal pairs) or learns a negative bias ($b$) to shift small positive interference below zero. The output ReLU then truncates this noise, effectively allowing the model to represent more features than dimensions by exploiting the non-linearity to filter out collisions.
> - We’ve added a brief explanation on this as a Footnote on page 3.
>
> While monosemanticity is typically defined for neurons, we overload the definition for features, experts, and models. See our background section for a more detailed description:
>
> > Monosemanticity is a characteristic of individual neurons, where a neuron's activation cleanly corresponds with a single $\alpha_i$ (i.e., features are basis-aligned). When features are orthogonal (not in superposition) but not basis-aligned, neurons remain *polysemantic* even though feature interference is minimal. In this paper, we focus on reducing superposition rather than enforcing basis-alignment. For brevity, when we describe models, experts, or features as ``more monosemantic," we mean they display less superposition.
> >
>
> We have revised the manuscript to include all of the above changes. Thank you for the detailed questions and feedback on the paper.

---

### Meta-Review · Area_Chair_acJx · 2026-01-09

**Summary:**

The reviewers raised consistent and serious concerns across soundness, clarity, and contribution. The consensus is that the paper relies too heavily on highly constrained toy models, with hidden dimensions as small as 1, and non-standard architectural choices that limit relevance to modern MoE systems.

**Reviewer Concerns:**

Concerns addressed:
- Terminology clarification (monosemantic neurons vs. features, interference vs. superposition).
- Notation and figure captions were partially clarified, especially for Sections 3–4.
- Overstatement of performance claims (e.g., “negligible” loss gap) was corrected to more cautious language.
- Rationale for omitting load balancing in some experiments was explained and generally accepted as a design choice (though not compelling to all).

Concerns outstanding:
- The work is incremental over prior analyses.
- Results from extremely small toy models are not transferable to real MoE architectures.
- The use of tied weights, output-ReLU placement, lack of SwiGLU, and absence of shared experts.
- Key claims about monosemanticity, feature counts, and interference are internally inconsistent or misleading.
- Clarity and formatting issues (including template non-compliance) persist.

**Reviewer Scores:**

The reviewers have engaged in the discussion.

---

### Decision · Program_Chairs · 2026-01-26

Reject